# Induction of WNT16 via Peptide-mRNA Nanoparticle-Based Delivery Maintains Cartilage Homeostasis

**DOI:** 10.3390/pharmaceutics12010073

**Published:** 2020-01-17

**Authors:** Huimin Yan, Ying Hu, Antonina Akk, Muhammad Farooq Rai, Hua Pan, Samuel A. Wickline, Christine T.N. Pham

**Affiliations:** 1Department of Medicine, Washington University School of Medicine, St. Louis, MO 63110, USA; hyan23@wustl.edu (H.Y.); yinghu@wustl.edu (Y.H.);; 2John Cochran Veterans Affairs Medical Center, St. Louis, MO 63106, USA; 3Department of Orthopaedic Surgery, Washington University School of Medicine, St. Louis, MO 63110, USA; rai.m@wustl.edu; 4Department of Cardiovascular Sciences, University of South Florida, Tampa, FL 33620, USA; huapan@usf.edu

**Keywords:** nanoparticle, mRNA delivery, osteoarthritis

## Abstract

Osteoarthritis (OA) is a progressive joint disease that causes significant disability and pain and for which there are limited treatment options. We posit that delivery of anabolic factors that protect and maintain cartilage homeostasis will halt or retard OA progression. We employ a peptide-based nanoplatform to deliver Wingless and the name Int-1 (WNT) 16 messenger RNA (mRNA) to human cartilage explants. The peptide forms a self-assembled nanocomplex of approximately 65 nm in size when incubated with WNT16 mRNA. The complex is further stabilized with hyaluronic acid (HA) for enhanced cellular uptake. Delivery of peptide-WNT16 mRNA nanocomplex to human cartilage explants antagonizes canonical β-catenin/WNT3a signaling, leading to increased lubricin production and decreased chondrocyte apoptosis. This is a proof-of-concept study showing that mRNA can be efficiently delivered to articular cartilage, an avascular tissue that is poorly accessible even when drugs are intra-articularly (IA) administered. The ability to accommodate a wide range of oligonucleotides suggests that this platform may find use in a broad range of clinical applications.

## 1. Introduction

Osteoarthritis (OA) is a progressive disease that causes significant pain and suffering and for which there are limited medical treatment options. Although effective disease-modifying OA drugs (DMOADs) are critically needed, none has successfully emerged in the clinic [1]. The reasons for this failure are multifold, including the fact that primary OA is a complex, multifactorial disease with incompletely understood pathogenesis [2].

Much interest has recently focused on the role of inflammation and inflammatory cytokines in the pathogenesis of OA [3]. It is posited that an imbalance between inflammatory (catabolic) and anabolic factors leads to cartilage degeneration, a hallmark of OA. Thus, blockade of catabolic cytokines such as tumor necrosis factor alpha (TNF-α) and interleukin 1 beta (IL-1β) has gained attention as potential therapy. Indeed, intra-articular (IA) administration of IL-1 receptor antagonist (IL-1ra) exhibits disease-modifying effects in a rodent model of OA [3]. However, in clinical studies, the administration of commercially available IL-1ra (Anakinra, Kineret^®^) had no therapeutic effect in established knee OA [4] and so far, only offers short-term improvement in pain and function if administered within the first month following knee injury (NCT00332254) [5]. Thus, novel approaches that promote and/or maintain joint homeostasis to mitigate OA progression are highly desirable.

Wingless and the name Int-1 (WNT) comprises an evolutionarily conserved family of glycoproteins that signal through different pathways. Beta (β)-catenin-dependent (or canonical) WNT signaling leads to the nuclear translocation of β-catenin and transcriptional activation of target genes that regulate many crucial aspects of cell fate and function during embryogenesis [6] Non-canonical, β-catenin-independent WNT signaling pathways and function on the other hand are less well understood [7]. The WNT signaling pathway has been implicated in the pathogenesis of OA [8]. WNT3a-dependent activation of the canonical β-catenin-dependent WNT signaling pathway stimulates catabolic activities, resulting in an OA-like phenotype [9,10]. In contrast, WNT16-deficiency leads to more severe OA in a rodent model, with increased chondrocyte apoptosis and decreased expression of lubricin [11], an essential joint lubricant [12]. In vivo injection of recombinant WNT16 in a Xenopus assay buffers the activation of canonical WNT3a [11]. Thus, we posit that overexpression of WNT16 antagonizes canonical WNT signaling to halt cartilage loss and mitigate the progression of OA.

The delivery of ribonucleic acids (RNA), small interfering RNA (siRNA), messenger RNA (mRNA), and microRNA (miRNA) into cells has been attempted using a variety of platforms, including lipid-based nanocarriers [13]. However, once taken up inside the cells, these particles are often trapped inside endosomes, with slow release of the RNA structures. Herein, we employ a cytolytic peptide, melittin, that has been modified to significantly attenuate its pore-forming capacity while maintaining its ability to insert into membrane bilayers [14,15,16], as well as improving its interactions with oligonucleotides [17,18]. Our previous work has shown that the modified peptide, called p5RHH, forms a self-assembled nanostructure that facilitates endosomal escape and rapidly delivers siRNA to the cytoplasm, to down-modulate specific gene expression in vitro and in vivo [17,18,19,20]. To our knowledge, the delivery of mRNA using p5RHH for WNT16 overexpression has not been reported. Herein, we present a proof-of-concept study showing that WNT16 mRNA can be efficiently delivered to articular cartilage and that its overexpression modulates cartilage homeostasis ex vivo.

## 2. Results

### 2.1. Hyaluronic Acid (HA) Coating Enhances Cellular Uptake of the Nanoparticle (NP)

HA, a naturally occurring and highly biocompatible linear polysaccharide that is a major component of the extracellular matrix (ECM), is present in high concentration in the synovial fluid and hyaline cartilage and binds to the surface receptor CD44 expressed on chondrocytes [21]. We posit that functionalization of the NPs with an HA coating will enhance cellular uptake through the interaction with CD44. Indeed, HA coating significantly enhanced cellular interaction and uptake of the NPs (Figure 1). We have previously shown that the self-assembled p5RHH-siRNA NPs deeply penetrate human cartilage to deliver the RNA load to chondrocytes [20]. In the present study we showed that the HA coating did not lead to increased NP interaction and accumulation in the superficial layer or prevent the penetration of the NPs into the deep zones of cartilage (Figure 2).

### 2.2. p5RHH-mRNA NP Preparation and Characterization

Next, we prepared the peptide-mRNA NPs by mixing a set amount of p5RHH peptide (10 μmol) with increasing concentrations of WNT16 mRNA (~1100 nucleotides, nt). The mixing of 10 μmol p5RHH with 1 μg of WNT16 mRNA (peptide:mRNA ratio 3500:1) yielded a NP of ~65 nm after application of the HA coating, as measured by transmission electron microscopy (TEM, Figure 3), and a zeta potential of ~30 mV by dynamic light scattering (DLS, Table 1). Increasing the concentration of mRNA resulted in a significantly increased particle diameter (>200 nm by TEM at an mRNA concentration of 4 μg and a peptide:mRNA ratio of 875:1, Figure 1A) and marked heterogeneity in the sizes of the NPs. The larger NP size measured by DLS (Table 1) suggests aggregates from smaller particles, which is supported by the TEM images (Figure 3, right panel). While DLS is a calculation that fits the light scattering data to an algorithm based on Mies scattering theory, TEM allows for direct visualization of the transfective particles and exclusion of the larger aggregates from the calculation, which we know, are not transfective from prior work [17,18].

### 2.3. Delivery of Enhanced Green Fluorescent Protein (eGFP) mRNA in Cartilage Explants

To begin exploring the efficacy/efficiency of our nanoplatform in the delivery of mRNA to express anabolic factors, we first used eGFP mRNA (~1000 nt) for ease of detection of the translated product. HA-coated p5RHH-eGFP mRNA NPs were prepared, as detailed in Materials and Methods and added to 5 mm^2^ cartilage discs from human OA knee joints. The cartilage explants were harvested after 48 h in culture and examined for eGFP expression. We observed efficient cartilage expression of eGFP with transfection of the NPs (Figure 4).

### 2.4. Delivery of WNT16 mRNA in Cartilage Explants

We next tested the delivery of WNT16 mRNA. The purified WNT16 mRNA construct was produced commercially and contained the appropriate endcaps and poly-A tail. HA-coated p5RHH-WNT16 mRNA NPs were prepared using 3 different concentrations of mRNA: 1 μg, 2 μg, or 4 μg (as detailed in Section 2.2). The self-assembled NPs were incubated with human cartilage explants for 48 h then examined for protein expression of WNT16, β-catenin, and WNT3a. We found that expression of WNT16 was significantly enhanced with the delivery of mRNA at 1 μg and 2 μg, but not at 4 μg (Figure 5A,B), likely because the size of the self-assembled NPs at this concentration (4 μg) of mRNA was too big for efficient cartilage penetration. Increased WNT16 expression was accompanied by decreased β-catenin (Figure 5C,D) and WNT3a (Figure 5E,F).

### 2.5. Effect of WNT16 Overexpression on Cartilage Homeostasis

We examined the downstream effects following the delivery of the NPs. We chose the NPs formulated with 1 μg of WNT16 mRNA, since this concentration resulted in the highest expression WNT16 (Figure 5A). We assessed the expression of lubricin, an essential joint lubricant that protects against chondrocyte apoptosis and cartilage deterioration [12]. We observed that WNT16 mRNA delivery led to a significant upregulation of lubricin-expressing cells and lubricin in the superficial layer of the cartilage explants (Figure 6A,B), which in turn led to suppression of chondrocyte apoptosis, as evidenced by a decrease in the number of terminal deoxynucleotidyl transferase-mediated dUTP nick end labeling (TUNEL)+ cells (Figure 6C,D).

## 3. Discussion

The challenge of delivering nanotherapeutics to cartilage in effective doses in vivo is well known [22]. Critical barriers include inefficient delivery to the chondrocytes residing in the avascular cartilage tissue and the dense ECM that excludes large particles from entering the deeper layers in order to deliver the therapeutic cargo. We have employed the amphipathic cationic peptide p5RHH that is a modified version of the natural peptide melittin, which rapidly forms a biocompatible and stable nanocomplex upon mixing of the peptide and nucleotide components [17,18,23]. The mechanism by which the modified melittin-derived peptide forms self-assembled nanostructures has been previously described [17,18]. In brief, modifications to p5RHH, with the addition of histidine and arginine moieties, enhance electrostatic interactions, permitting formation of noncovalent hydrogen bonds between oligonucleotides and the peptide [24]. The complex protects the RNA from degradation and once taken up inside the cell, the peptide can facilitate endosomal escape and coordinated release of nucleic acid structures into the cytoplasm [18]. Herein we show for the first time that p5RHH can also complex with mRNA structures (up to ~1100 nt) to form stable NPs of ~65 nm, small enough to penetrate cartilage for delivery and translation of the mRNA. An HA coating further enhances cellular uptake without retarding the ability of the NPs to deeply penetrate human cartilage. The versatility of the platform to incorporate short and long nucleotide structures significantly broadens the range of clinical applications for this technology.

Approaches to OA treatment have recently shifted toward anabolic pathways that promote cartilage repair and homeostasis. Fibroblast growth factors (FGFs) are important regulators of cartilage development and homeostasis [25]. IA injection of FGF-18 in a rat meniscal tear model induces new cartilage formation [26]. Sprifermin (AS902330), a recombinant form of human FGF-18 injected IA in patients with advanced or end-stage OA shows early promise; however, durability of response at two years was uncertain [27]. Likewise, excessive (β-catenin-dependent) canonical WNT activation leads to cartilage breakdown and increases risk of OA [8,28,29]. A small molecule inhibitor of the WNT pathway (SM04690) shows protective and regenerative effects in an OA animal model [30] and has the potential to be disease modifying in knee OA; however, long-term effects are still unknown (ongoing trials NCT03727022). In the present study, we show that overexpression of WNT16 suppresses canonical β-catenin/WNT3a signaling. We envision that the p5RHH platform, by its ability to accommodate a wide range of oligonucleotide structures (siRNA, mRNA, and others) without the need for backbone or end-piece alterations, will enable the delivery of a “cocktail” of factors (anti-inflammatory and anabolic) that should control cartilage loss and maintain homeostasis, mitigating OA progression.

In conclusion, we have shown that melittin-derived p5RHH peptide self-assembles with mRNA to form stable nanostructures that deeply penetrate cartilage for efficient expression of WNT16 ex vivo. These results hold promise that this approach will overcome the shortcoming of slow RNA release encountered by lipid-based NPs [13]. In future studies, we will test the effectiveness of WNT16 overexpression in maintaining cartilage homeostasis in vivo.

## 4. Materials and Methods

### 4.1. Preparation of HA-Coated p5RHH-mRNA NPs

Ten milligrams of sodium hyaluronic acid (part# HA1M-1, Lifecore Biomedical, Chaska, MN, USA) was dissolved in 1 mL HBSS with Ca++/Mg++ by sonification for 60 min and ultracentrifuged at 90,000g for 40 m. The supernatant was aliquoted and stored at −80 °C until use. p5RHH peptide (VLTTGLPALISWIRRRHRRHC, provided by Genscript, Piscataway, NJ, USA) was dissolved at 10 mM in DNase-, RNase-, and protease-free sterile purified water (Cellgro at Corning, Tewksbury, MA, USA) and stored in 10 μL aliquots at −80 °C until use.

The p5RHH-Cy3-labeled siRNA NPs were prepared as previously described [20,31]. The p5RHH-mRNA NPs were prepared as follows: 1 μg of Cy5-eGFP mRNA (TriLink Biotechnologies, San Diego, CA, USA), or 1, 2, or 4 μg of WNT16 mRNA (TriLink Biotechnologies) in HBSS with Ca++/Mg++ were added to 10 μmol of p5RHH peptide (in a total volume of 100 μL), mixed well, and incubated at 37 °C for 40 min. After incubation, 5 μL of HA was added to the self-assembled NPs and placed on ice for 5 min. This mixture was diluted into a total volume of 500 µL with culture medium for in vitro transfection. NP size was measured by TEM and zeta potential by DLS. To calculate the actual spherical volume of the NPs from their “flattened” shape acquired during the TEM drying process, we used the formula for the volume of a right cylinder V = πr^2^h, where V = volume, r = radius, and h (height). Height was assumed to be 1/5th of their flattened diameter. The radius for a sphere of the same volume as the right cylinder was calculated from the formula V = 4/3πr^3^.

### 4.2. Human Cartilage Explant Culture

Human cartilage explants were obtained from patients, through a protocol (ID # 201104119 approved 01/10/2019) approved by the Washington University in St. Louis Institutional Review Board (IRB), at the time of total knee arthroplasty. All study participants provided written informed consent. The de-identified cartilage tissues were washed several times with HBSS containing antibiotics, then incubated in Dulbecco’s Modified Eagle Medium/Nutrient Mixture F-12 (DMEM/F12) (1:1) medium containing 10% fetal bovine serum (FBS), penicillin/streptomycin (100 U/0.1  mg/mL), amphotericin B (0.25  µg/mL), and ciproflaxin (10  μg/mL) in a 6-well plate at 37 °C and 5% CO_2_ for 2–3 days. The explants were then transferred to a 96-well plate and subsequently exposed to the aforementioned p5RHH-mRNA NPs for 48  h. The excess NPs were washed off after 48  h incubation. The cartilage explants were harvested and then embedded in Tissue-Tek optimal cutting temperature (O.C.T.) compound (Sakura Finetek, Torrance, CA, USA) and sectioned for analysis.

### 4.3. In Vitro NP Uptake by Bone-Marrow-Derived Macrophages (BMMϕ)

All animal experiments were performed in compliance with guidelines and protocols approved by the Division of Comparative Medicine at Washington University in St. Louis. The animal protocol (Animal Welfare Assurance # A-3381-01, approved 02/28/2019) is subjected to annual review and approval by The Animal Studies Committee of Washington University. Bone marrow from C57BL/6 WT mice (Cat# 000664, Jackson Laboratory, Bar Harbor, ME, USA) was cultured in complete RPMI-1640 medium with 10% fetal bovine serum containing recombinant murine granulocyte-macrophage colony stimulating factor (GM-CSF) (10 μg/mL, Cat# PMC2015, Thermo Fisher Scientific, Waltham, MA, USA) for 7 days at 37 °C. Cultured cells were plated in 12-well plates at 0.5  ×  10^6^ cells/well overnight. The cells were starved for 30 min prior to stimulation with 10 µg/mL lipopolysaccharide (LPS) (Cat# L2762, Sigma-Aldrich, St. Louis, MO, USA) for 15 min. The cells were subsequently cultured in Opti-Minimum Essential Medium (MEM) (Thermo Fischer Scientific) containing HA-coated Cy3-labeled NPs or uncoated Cy3-labeled NPs at 37 °C for the indicated times. The cells were collected with ethylenediaminetetraacetic acid (EDTA) solution (1:10 dilution with phosphate buffered saline (PBS), spun down, resuspended in Flow Cytometry Staining (FACS) buffer, and analyzed by flow cytometry. For confocal analysis, cells were cultured and fixed with 4% paraformaldehyde in PBS, permeabilized with 0.1% Triton X-100/PBS, and blocked with 8% BSA. F-actin (Cat# T7471, 1:200 dilution, Invitrogen at Thermo Fischer Scientific) was added and the cells mounted with VECTASHIELD containing 4’, 6-diamidino-2-phenylindole (DAPI) (Cat# D3571 Molecular Probes at Thermo Fischer Scientific). The images were captured by a ZEISS LSM 880 confocal laser scanning microscope.

### 4.4. TUNEL Assay

The apoptotic assay was performed to identify DNA fragmentation associated with terminal deoxynucleotidyl transferase-mediated (dUTP) nick end labeling (TUNEL). Detection of apoptotic cells was performed on non-fixed frozen cartilage sections using an in situ cell death detection kit (Cat#: 11–684-795–910, Roche at Millipore Sigma, St. Louis, MO, USA). In brief, the cartilage sections were rinsed with PBS, then permeabilized with 0.5% TWEEN-20/PBS for 15 min, and blocked with 8% BSA solution. Freshly prepared TUNEL reaction mixture, according to the manufacturer’s protocol, was applied to the sections for 1 h at 37 °C, rinsed 3 to 5 times with PBS, and probed with COL2 antibody (1:200 dilution, generously provided by L. J. Sandell and M. F. Rai, Washington University, St. Louis), followed by tetramethylrhodamine (TRITC)-conjugated anti-rat secondary antibody (1:100 dilution, Cat# 712-295-153, Jackson Immuno Research, West Grove, PA, USA), and counterstained with DAPI (1:1000 dilution, Vector Laboratories, Burlingame, CA, USA). The sections were mounted with VECTASHIELD mounting medium with DAPI (Cat#: H-1200, Vector Laboratories). The TUNEL^+^ cells were enumerated across non-overlapped fields. Data represent 6–8 sections per cartilage and 4–6 patients per treatment.

### 4.5. Confocal Microscopy

After incubation with eGFP mRNA NPs for 48 h, the cartilage explants were harvested and sectioned. Frozen sections (9 μm) were rinsed, fixed, and covered with VECTASHIELD mounting medium with DAPI (1:1000, Vector Laboratories) at room temperature. The images were acquired with a ZEISS LSM 880 confocal laser scanning microscope—15 to 20 cells per section and 3–4 sections were analyzed with the software ZEN. The data was presented as the mean fluorescent intensity per cell.

### 4.6. Immunohistochemistry

Formalin-fixed, O.C.T-embedded 9 μm sections of human cartilage explants were probed with WNT16 (1:100 dilution, Cat# LS-A9629, LifeSpan Biosciences, Seattle, WA, USA), WNT3A (1:100 dilution, Cat# OABF00803, Aviva Systems Biology, San Diego, CA, USA), Lubricin (1:200 dilution, Cat# 55463, MP Biomedicals, Irvine, CA, USA), or β-catenin (1:100, Cat# ab16051, Abcam, Cambridge, MA, USA) at room temperature for 1 h. After washing, the sections were incubated with the corresponding horse radish peroxidase (HRP)-conjugated secondary antibodies for 1 h. Data presented were derived from 6–8 cartilage sections. The pattern was confirmed on 4–6 independent human cartilage explants.

### 4.7. Statistics

Comparisons between two groups were performed by Student’s *t*-test, and between multiple groups (≥3) by one-way ANOVA followed by Bonferroni’s correction for multiple comparisons. Differences between experimental groups at a *p* value of <0.05 were considered significant.

## Figures and Tables

**Figure 1 pharmaceutics-12-00073-f001:**
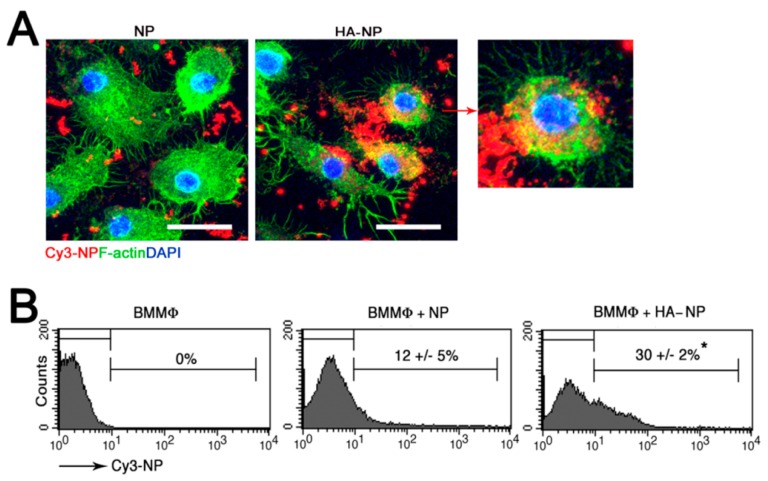
Cy3-labeled p5RHH naked nanoparticle (NP) or hyaluronic acid-coated p5RHH NP (HA-NP) were incubated with bone-marrow-derived macrophages (BMMϕ) for 4 h then analyzed by confocal microscopy (**A**). NP = red; F-actin = green; scale bar = 25 µm. HA-coating enhanced cellular interaction and uptake (higher magnification: far right panel). (**B**) Flow cytometric analysis of BMMϕ after 4 h of incubation with NPs. Values represent mean ± SEM, n = 3, * *p* < 0.05.

**Figure 2 pharmaceutics-12-00073-f002:**
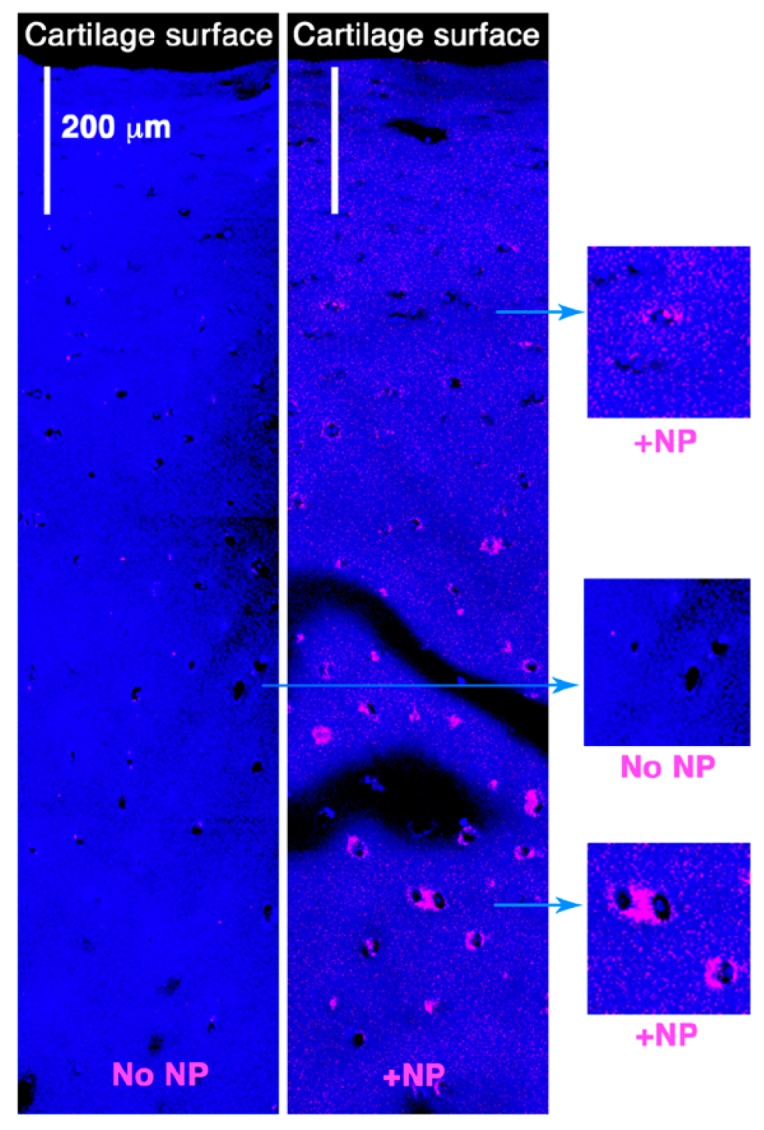
Cy3-labeled HA-NPs (~55 nm) were incubated with 5 mm^2^ cartilage discs from human OA knee. After 48 h, cartilage explants were washed, processed, and examined for depth of NPs penetration by confocal microscopy. The HA-NPs deeply penetrated the cartilage explants (up to ~1 mm in depth).

**Figure 3 pharmaceutics-12-00073-f003:**
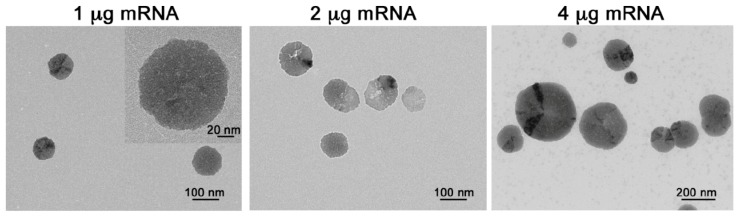
HA-coated p5RHH-WNT16 mRNA NPs were generated by mixing 10 μmol of p5RHH with 1 µg mRNA (peptide:mRNA ratio 3500:1), 2 µg of mRNA (peptide:mRNA ratio of 1750:1), or 4 µg of mRNA (peptide:mRNA ratio 875:1). Inset (in the left panel) shows the NP at a higher magnification.

**Figure 4 pharmaceutics-12-00073-f004:**
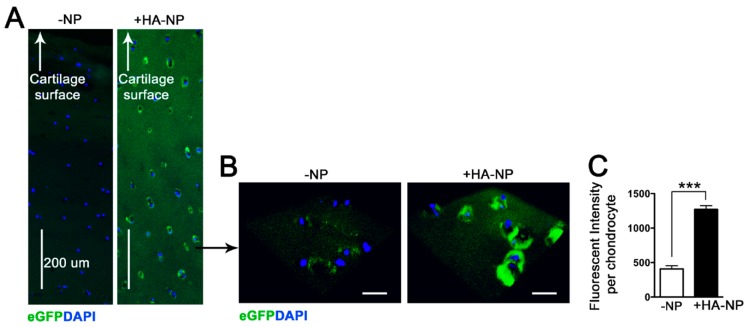
HA-coated p5RHH-eGFP mRNA NPs at 1 µg of mRNA and 10 µmol of p5RHH were incubated with 5 mm^2^ cartilage discs derived from human OA knee joints. After 48 h, cartilage explants were washed, processed, and examined for eGFP expression (green). (**A**) eGFP expression was detected in all layers of cartilage. (**B**) Higher magnification revealed eGFP expression in chondrocytes residing in the deep zone of cartilage. DAPI stains nuclei blue. Scale bar = 25 µm. (**C**) Mean eGFP fluorescent intensity ± SEM per chondrocyte (n = 20 chondrocytes assessed per treatment condition). N = 3 cartilage discs per treatment. Scale bar = 25 µm, *** *p* < 0.001.

**Figure 5 pharmaceutics-12-00073-f005:**
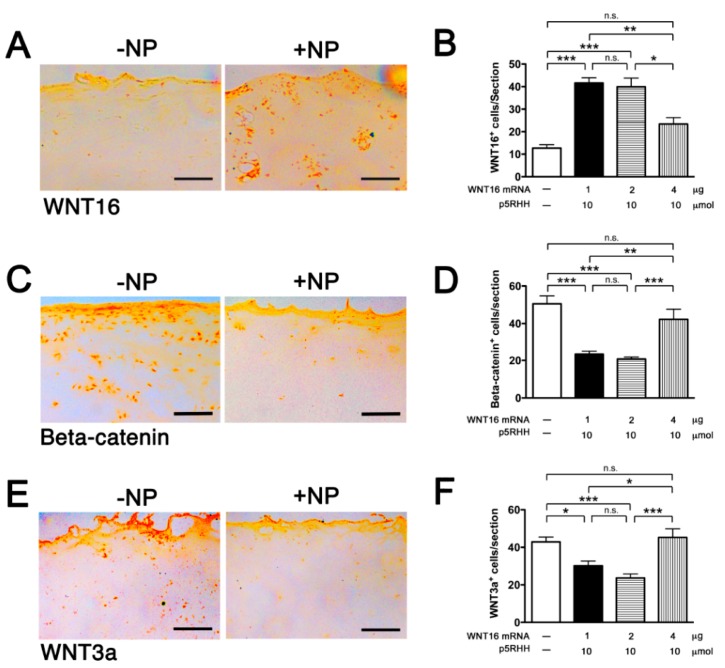
HA-coated p5RHH-WNT16 mRNA NPs generated at the indicated concentrations of mRNA and p5RHH were incubated with 5 mm^2^ cartilage discs derived from human OA knee joints. After 48 h, cartilage explants were washed, processed, and examined for WNT16 (**A**,**B**), Beta-catenin (**C**,**D**), and WNT3a (**E**,**F**) expression. Immunohistochemistry (IHC) photomicrographs were derived from cartilage discs transfected with 1 μg of peptide-mRNA NPs. Scale bar = 100 μm. The numbers of WNT16+, beta-catenin+, and WNT3a+ cells/cartilage section were enumerated. Values represent mean ± SEM. Data were derived from 6 to 8 cartilage sections, from 4–6 independent human cartilage explants. * *p* < 0.05, ** *p* < 0.01, *** *p* < 0.001, n.s. = not significant.

**Figure 6 pharmaceutics-12-00073-f006:**
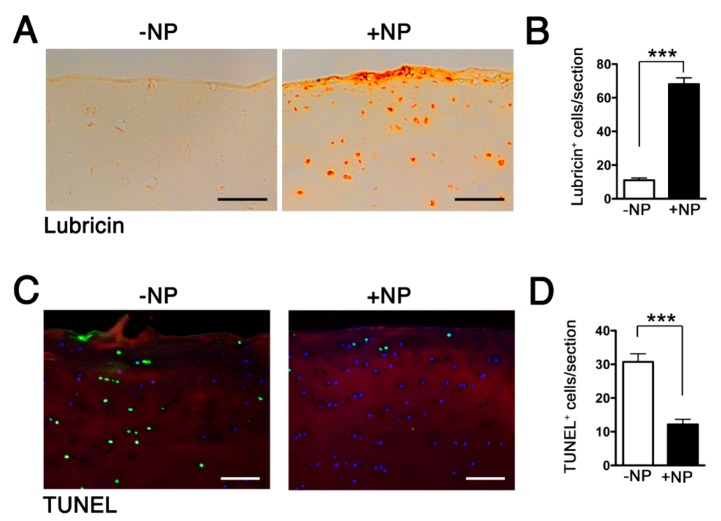
HA-coated p5RHH-WNT16 mRNA NPs (generated at 1 μg of mRNA and 10 μmol of p5RHH) were incubated with 5 mm^2^ cartilage discs derived from human OA knee joints. After 48 h, cartilage explants were washed, processed, and examined for lubricin expression (**A**,**B**) and terminal deoxynucleotidyl transferase and nick-end labeling (TUNEL)+ cells (**C**,**D**). Scale bar = 100 μm. The numbers of Lubricin+ and TUNEL+ cells/cartilage section were enumerated. Values represent mean ± SEM. Data were derived from 6–8 cartilage sections, from 4–6 independent human cartilage explants. *** *p* < 0.001.

**Table 1 pharmaceutics-12-00073-t001:** Characteristics of NPs at various peptide:mRNA ratios.

Particle Composition Peptide:mRNA (mol:mol)	3500:1	1750:1	875:1
Average size (nm) TEM	64.78 ± 8.344 (n = 50)	105.5 ± 6.959 (n = 150)	205.1 ± 6.109 (n = 500)
Average size (nm) DLS	181	177	3000
Zeta potential (mV)	−30.06 ± 0.82	−28.79 ± 1.77	−31.48 ± 2.59

NP size was assessed by DLS and TEM and calculated according to the formula detailed in the materials and methods section. n = number of NPs assessed per condition.

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
