# Peer review of "Induction of WNT16 via Peptide-mRNA Nanoparticle-Based Delivery Maintains Cartilage Homeostasis"

_pharmaceutics, 2020, doi:10.3390/pharmaceutics12010073_

Round 1
Reviewer 1 Report
In the manuscript, the authors employed a peptide-based nanoplatform to deliver WNT16 mRNA to human cartilage explants. They demonstrated that the delivery of the nanocomplex antagonized canonical b-catenin/WNT3a signaling and thus led to increased lubricin production and decreased chondrocyte apoptosis. The conclusions can be supported by the results. This nanoplatform can be further developed to other type of tissues. The manuscript can be accepted for publication after major revision by addressing the following comments. The writing of the manuscript also needs to be further revised.
In the introduction part, the current methods of using nanoparticles should be introduced and discussed to indicate the reason of using peptide particles as the carrier. The reason of coating HA to the particles should be discussed in the introduction part or at the beginning of section 2.1. Page 2, about the ratio of the peptide and mRNA, does it refer to mol ratio? Why with the increase of the mRNA content from 1 to 4 μg, the peptide:mRNA ratio was increased from 1:3500 to 1:875? Actually, the ratio should be decreased. What is the mechanism of the formation of self-assembled nanocomplex of the peptide and mRNA? In Figure 1, the authors stated that the scale bars in figure 1A are 100 μm but the sizes of the nanoparticles were within nanometer range. This mistake should be revised. The abbreviations should be defined at the first time showing in the manuscript. It is difficult to understand the meaning of the abbreviations. For the delivery of mRNA in cartilage explants, which location of the explants was used to measure the fluorescence intensity since the nanoparticles could penetrate the explants? The y-axis of Figure 4B should be “fluorescence intensity”. What are the conclusions? A paragraph is needed to give a conclusion of the study.Author Response
Dear Editors and Reviewers,
We would like to thank the Editors and Reviewers for their comments and critiques. We have addressed them in the manuscript and below. We hope that the manuscript is now acceptable for publication in Pharmaceutics.
Reviewer 1
In the introduction part, the current methods of using nanoparticles should be introduced and discussed to indicate the reason of using peptide particles as the carrier.
Response: Done (see Introduction)
The reason of coating HA to the particles should be discussed in the introduction part or at the beginning of section 2.1
Response: Done (see new section 2.1)
Page 2, about the ratio of the peptide and mRNA, does it refer to mol ratio?
Response: Yes, this is now specified (mol:mol) in Table 1
Why with the increase of the mRNA content from 1 to 4 μg, the peptide:mRNA ratio was increased from 1:3500 to 1:875? Actually, the ratio should be decreased.
Repsonse: We thank the Reviewer for pointing out these confusing ratios. The ratios are indeed decreased if expressed as peptide:mRNA. The correct ratios are listed in Table 1.
What is the mechanism of the formation of self-assembled nanocomplex of the peptide and mRNA?
Response: Modifications to p5RHH, with the addition of histidine and arginine moieties, enhance electrostatic interactions, permitting formation of noncovalent hydrogen bonds between oligonucleotides and the peptide. This is now added to the Discussion with references.
In Figure 1, the authors stated that the scale bars in figure 1A are 100 μm but the sizes of the nanoparticles were within nanometer range. This mistake should be revised.
Response: We thank the Reviewer for pointing out the mistake. This has been corrected.
The abbreviations should be defined at the first time showing in the manuscript.
Response: We have made every attempt to define abbreviations at first use.
For the delivery of mRNA in cartilage explants, which location of the explants was used to measure the fluorescence intensity since the nanoparticles could penetrate the explants?
Response: Fluorescent intensity represents average fluorescence per chondrocyte. We have corrected the y-axis label to figure 4C. We have included a lower magnification image of the cartilage explant in figure 4A to show that eGFP expression can be seen throughout all layers.
What are the conclusions? A paragraph is needed to give a conclusion of the study.
Response: A concluding paragraph has been added.

Reviewer 2 Report
Authors describe the fabrication of peptide NPs complexed with WNT16-RNA, and check if they could be suitable for cartilage regeneration. The manuscript is well written. However, I consider that some parts should be explained more, as they are just too brief.
Introduction: The part describing WNT and WNT16 should be explained in more detail. And cite if there are other authorsthat have encapsulated this RNA with similar NPs, even if they are for different applications. In the objective the hyaluronic should also be mentioned, ,as it can also play an important role in targeting the NPs. Materials and methods: Measuring the size of the NPs by DLS is necessary, to know the hydrodynamic diameter, instead of the maths carried out through the TEM images. And, which was the n used for the calculus of the size by TEM. Results: Legend in figure 1. I think there is a typo and it is 100nm not um. Table 1. What is the stander deviation of the zeta potential? In figure 2, I would add another sample, that would be NPs non coated with HA, to ensure that the NPs can cross better the cartilage when decorated. And also, I am surprise that around 60 nm can be observed on the NPs, I would think they are aggregating. Thus, I would suggest to repeat the experiment with more magnification, to see what is there. And, it is true that the NPs when decorated with the hyaluronic seem to interact more with the cells (according to FACS). But, the images should be improved, to see the localization, if they are just interacting with the membrane or are really inside the cells. I just don’t see it clear that they are interacting with the confocal image. And, considering that they are using macropages, that they normally uptake NPs easily, I am surprise with the images. This should explained better and better images of confocal should be added. Why the authors used some marrow derived macrophages? The section 2. Delivery of eGFP mRNA in cartilage explants should be in the methods part. And the results of this commented in this section.Author Response
Responses to Reviewer 2
Dear Referee,
Many thanks for your time and valuable comments! We have revised our manuscript according to the review reports step by step. Please check it.
Thanks again!
Introduction: The part describing WNT and WNT16 should be explained in more detail.
Response: More details on WNT have been added to the Introduction
And cite if there are other authors that have encapsulated this RNA with similar NPs, even if they are for different applications.
Response: To our knowledge no other publication has previously described encapsulated WNT16 mRNA NP.
In the objective the hyaluronic should also be mentioned, as it can also play an important role in targeting the NPs.
Response: HA coating is now expanded under subheading 2.1.
Materials and methods: Measuring the size of the NPs by DLS is necessary, to know the hydrodynamic diameter, instead of the maths carried out through the TEM images. And, which was the n used for the calculus of the size by TEM.
Response: DLS measurements are now included in Table 1. Note the difference in size between DLS and TEM. The significantly larger value from DLS can be attributed to interparticle interaction, leading to aggregates. It has been shown that even a small number of larger particles (1-2 vol %) can significantly change the particle size distribution, as DLS cannot differentiate between small aggregates and larger particles (Caputo et al. J Control Release 2019, 299:31-43). DLS also is a calculation that fits the light scattering data to an algorithm based on Mies scattering theory. TEM on the other hand allows for direct visualization of the transfective particles and exclusion of the larger aggregates from the calculation, which we know are not transfective from prior work (Hou et al Biomaterials 2013, 34:3110-3119; Hou et al. ACS Nano 2013, 7:8605-8615). The TEM calculations complement the results obtained from DLS. The n used to calculate the average size of NP by TEM has now been included in Table 1.
Results: Legend in figure 1. I think there is a typo and it is 100nm not um.
Response: Corrected
Table 1. What is the stander deviation of the zeta potential?
Response: Included in Table 1
In figure 2, I would add another sample, that would be NPs non coated with HA, to ensure that the NPs can cross better the cartilage when decorated.
Response: We are not claiming that the NP can penetrate cartilage better with an HA coating. Rather, we want to show that HA-coated NP does not get hung up in the upper layers due to interaction with chondrocytes in this zone. We show that HA-coated NP can penetrate to the deeper layers of cartilage to about the same extent as previously shown with naked NP (Yan et al. Proc Natl Acad Sci USA 2016, 113, E6199-E6208
And also, I am surprise that around 60 nm can be observed on the NPs, I would think they are aggregating. Thus, I would suggest to repeat the experiment with more magnification, to see what is there.
Response: There are some aggregates, especially at the 4 ug concentration of WNT16 mRNA (Fig. 3). A higher magnification of the particle is now included in fig. 3 (at 1 ug WNT mRNA)
And, it is true that the NPs when decorated with the hyaluronic seem to interact more with the cells (according to FACS). But, the images should be improved, to see the localization, if they are just interacting with the membrane or are really inside the cells. I just don’t see it clear that they are interacting with the confocal image. And, considering that they are using macropages, that they normally uptake NPs easily, I am surprise with the images. This should explained better and better images of confocal should be added.
Response: HA-coating increased both interaction and uptake (higher magnification confocal images are now included). Although macrophages normally take up NP easily as stated by the Reviewer, we noted a difference at the 4-hour time point. Shorter incubation time did not result in significant uptake, without and with HA coating (data not shown).
Why the authors used some marrow derived macrophages?
Response: We used bone marrow derived macrophages since they readily take up NP while primary chondrocytes are relatively inefficient at phagocytosis.

Reviewer 3 Report
The authors present an interesting concept, which is relevant to current theranostics. However, several deep flaws exist which must be covered before the study could be potentially published. I have indicated the relevant points that need to be worked on in the pdf of the manuscript attached below.

Author Response
Responses to Reviewer 3
Dear Referee,
Many thanks for your time and valuable comments! We have revised our manuscript according to the review reports step by step. Please check it.
Thanks again!
What is the size of these particles? Why are all sizes not shown here?
Response: The points we are trying to make with the HA coating are:
HA coating increases cellular interaction and uptake of NP of the same size (55 nm in this case) HA coating does not cause NP to be hung up in the superficial layer due to interaction with chondrocytes in this zone. We are not claiming that HA coating enhances NP depth of penetration
Accumulation at the surface of the explants is not seen…so where did they go?
Response: In figure 2, the cartilage explants were incubated with Cy3-labeled NP for 48 hours then the excess NP washed off prior to processing and imaging. Cy3-labeled NP can be seen penetrating all layers of cartilage. In figures 4, 5 & 6, we did not use labeled NP. Only the downstream products (eGFP, WNT16 proteins) were examined.
Wnt3a is higher than control!
Response: There is no statistical difference between control and NP at 4ug of WNT16 mRNA
Why are the other sample concentrations not tried and reported here?
Response: We used the NP made with 1 ug of WNT16 mRNA since this concentration gave the highest expression of WNT16

Round 2
Reviewer 1 Report
The authors have addressed the questions. In the revised version, Figure 1 is overlapped with another image. The manuscript can be accepted for publication after the Figure 1 is revised or corrected.
Reviewer 2 Report
All my comments have been addressed adequately.
Reviewer 3 Report
The responses from the authors is in agreement with the previous comments in review-1. The manuscript can now be taken ahead for finalization and publication.